# The Socio-Environmental Determinants of Childhood Malnutrition: A Spatial and Hierarchical Analysis

**DOI:** 10.3390/nu16132014

**Published:** 2024-06-25

**Authors:** Austin Sandler, Laixiang Sun

**Affiliations:** 1Department of Agricultural and Resource Economics, University of Connecticut, 1376 Storrs Rd., Unit 4021W.B. Young 302, Storrs, CT 06269, USA; austin.sandler@uconn.edu; 2Department of Geographical Sciences, University of Maryland, 2181 LeFrak Hall, 7251 Preinkert Dr., College Park, MD 20740, USA; 3School of Finance & Management, SOAS University of London, London WC1H 0XG, UK

**Keywords:** child malnutrition, wasting, stunting, Kenya, Nigeria, DHS, hierarchical, spatial

## Abstract

Despite a remarkable reduction in global poverty and famines, substantial childhood malnutrition continues to persist. In 2017, over 50 million and 150 million young children suffered from acute malnutrition (*wasting*) and chronic malnutrition (*stunting*), respectively. Yet, the measurable impact of determinants is obscure. We evaluate proposed socio-environmental related determinants of stunting and wasting across Kenya and Nigeria and quantify their effectiveness. We combine health and demographic data from Kenya and Nigeria Demographic Health Surveys (2003, 2008–2009, 2013, 2014) with spatially explicit precipitation, temperature, and vegetation data. Geospatial and disaggregated data help to understand better who is at risk and where to target mitigation efforts. We evaluate the responsiveness of malnutrition indicators using a four-level random intercept hierarchical generalized logit model. We find that spatial and hierarchical relationships explain 28% to 36% of malnutrition outcome variation. Temporal variation in precipitation, temperature, and vegetation corresponds with more than a 50% change in malnutrition rates. Wasting is most impacted by mother’s education, family wealth, clinical delivery, and vaccinations. Stunting is most impacted by family wealth, mother’s education, clinical delivery, vaccinations, and children asymptomatic of fever, cough, or diarrhea. Remotely monitored climatic variables are powerful determinants, however, their effects are inconsistent across different indicators and locations.

## 1. Introduction

Childhood malnutrition is a pernicious public health issue. It is a detrimental and significant plight responsible for 45% of all deaths among children worldwide [1]. Malnutrition not only increases child morbidity and mortality, it also inhibits cognitive, social, and financial potential [2,3]. Ongoing progress to reduce malnutrition has so far been insufficient to attain the World Health Assembly targets for 2025 and the Sustainable Development Goals for 2030 (i.e., a 40% reduction in stunting prevalence and reduce wasting prevalence to less than 5% by 2025, and by 2030 end all forms of malnutrition) [4,5]. Despite downward global trends, only 26 of 202 countries are on track to meet the undernutrition target [6,7].

Causes of child malnutrition are broadly divided into two etiological categories: illness-related or non-illness-related [8]. The focus of this paper is to evaluate the latent determinants that impact the severity and variability of non-illness-related childhood malnutrition. Non-illness-related malnutrition stems from economic, social, environmental, political, or cultural factors that decrease nutrient intake and negatively affect growth and development [9]. Malnutrition severity is measured by deterioration in various key anthropometric indicators.

Two of the most widely studied indicators are *wasting* and *stunting*. Wasting indicates a deficit in tissue and fat mass, either from weight loss or inability to gain weight. A child, aged 0 to 59 months, is defined as wasted if their weight-for-height is below negative two standard deviations from the median of the WHO Child Growth Standards [10]. Stunting indicates impeded skeletal growth. It is a measure of linear growth, representing chronic malnutrition accumulated over time. A child, aged 0 to 59 months, is defined as stunted if their height-for-age is below negative two standard deviations from the median of the World Health Organization Child Growth Standards [10].

Since the introduction of the 1990 UNICEF conceptual framework, it has been the standard for modeling the broad causes of child malnutrition. Driven in part by demand from governmental and non-profit aid agencies to successfully carry out their missions, there has been an upsurge in studies attempting to corroborate the UNICEF framework with empirical evidence and adapt it to fit new paradigms [4,10,11,12,13,14,15,16,17,18,19,20,21,22,23,24,25,26,27,28,29,30,31]. Empirical evidence for non-illness-related childhood malnutrition determinants and policy interventions spans micronutrient supplementation [32], various sociodemographics [33,34,35], and climactic factors [36]. Other conceptualizations focus on different factors such as food security [37,38,39,40,41,42], risk factors [1,8,32,43,44,45,46,47,48,49], national economic growth [50,51], spatial composition [52,53,54], and utility maximization [55,56], all disseminating from a wider historical epistemology.

The UNICEF conceptual framework models child malnutrition as a hierarchical system to motivate the drivers of malnutrition [57]. The hierarchical strata include *immediate*, *underlying*, and *basic* classifications. Some specifications equate these strata to individual, household, and societal levels, whereby factors at one level influence other levels [29], while other specifications focus on distinguishing between *proximal* and *distal* determinants [58,59].

For example, in their report on the aggregate cross-county determinants of malnutrition, Smith and Haddad [2] identify specific subcategories of the UNICEF framework. They specify dietary intake and health status as the immediate determinants, which are influenced by the underlying determinants of food security (per capita national food availability), care for mothers and children (women’s education and women’s status relative to men’s), and health environment quality (safe water access), which are in turn influenced by the basic determinants of economic resource availabilities (per capita national income) and the political environment (democracy score). 

Despite long-observed environmental effects [60] and widely anticipated links between climate change and child malnutrition, evidence regarding the nature, severity, and variability of this relationship is just beginning to emerge across spatial and temporal scales [61]. Others have found that much of the evidence for the impact of climate on childhood malnutrition is based on a few heterogeneous studies with flawed methodologies [36]. Indeed, there exists abundant opportunities in the literature for studies with wider geographic coverage and greater attention to scale. Such studies should include multiple dimensions of nutrition outcomes and quantify the spatial, social, political, climatic, and economic determinants of malnutrition. 

Globally Nigeria has the second highest number of stunted children behind India. There are over 4.8 million wasted children and over 10 million stunted children in Nigeria. And Kenya provides a measure of external validity and contrast to this analysis, with over 278 thousand wasted children and over 1.8 million stunted children [10]. Including both countries adds variability in terms of malnutrition prevalence rates, governance, climate, population, economy, and culture. Kenya and Nigeria both have available health survey data with extensive temporal and spatial coverage that overlaps with the available geolocated historical climate data. Despite similar efforts in the literature in both Kenya [39,62,63,64] and Nigeria [11,12,65,66,67,68], this paper aims to be the first study to identify and quantify the non-illness-related and climatic determinants of stunting and wasting across Kenya and Nigeria through a spatially explicit hierarchical modeling approach consistent with the UNICEF [29,69] conceptual framework. 

## 2. Materials and Methods

### 2.1. Health Survey Study Design

The Demographic and Health Surveys (DHS) remain the most ubiquitous resource of its kind, with more than 350 surveys in over 90 countries across 30 years [70]. To supply the primary data on child health and household characteristics, we employ the Demographic and Health Surveys (DHS) Kids Recode files and the Geographic Data files for Kenya and Nigeria of DHS-IV (1997 to 2003), DHS-V (2003 to 2008), DHS-VI (2008 to 2013). The sample includes 48,086 Nigerian children and 28,421 Kenyan children. 

The sampling procedure is a multistage probability sample design, drawn from the sampling frame of the most recent census. Regions, zones, or provinces stratify national populations, and states or counties stratify regions. The final stratum contains a subpopulation from which clusters are randomly sampled. The extent of clusters vary; they can be a city block or apartment building in urban areas whilst being a village or group of villages in rural areas. Generally, geographic regions and urban or rural areas within each region partition the stratified samples of the Demographic and Health Surveys [71].

For example, the sampling frame for Nigeria partitions the country into 6 geographical regions, 36 states and the Federal Capital Territory, 774 local government areas, 8812 wards, and 665,000 census enumeration areas, each containing 48 households on average. The sample design for Nigeria DHS-VI selected 893 wards with a selection probability proportional to its population and stratified across urban and rural local government areas in each state. The sample design then selects 904 census enumeration areas from within the 893 wards and if a selected enumeration area contains less than 80 households, a neighboring enumeration area is added to form the primary sampling units or clusters. Finally, the sample design selects a fixed number of 45 households from each cluster to determine who to interview.

The probability of selecting a household is the probability of selecting the cluster multiplied by the probability of selecting the household within the cluster. Kenya DHS clusters all have 25 households whereas Nigeria DHS-VI clusters have 45 households; Nigeria DHS-V clusters have 41 households; and Nigeria DHS-IV clusters have 22 households on average. The number of clusters do not remain constant across time and space. Kenya DHS-V and DHS-IV have 400 clusters, whereas Kenya DHS-VI has 1612 clusters and Nigeria DHS-IV has 365 clusters; Nigeria DHS-V has 888 clusters; and Nigeria DHS-VI has 904 clusters. The stratified samples produce homogeneity within groups and heterogeneity between groups.

### 2.2. Data Measures

We employ data from three of four main questionnaires of the Demographic and Health Surveys. The Household Questionnaire characterizes the household in terms of physical amenities and a roster of the members of the household. The Biomarker Questionnaire characterizes the anthropometric measurements and biochemical indicators of eligible members of the household. The Woman’s Questionnaire contains a birth history roster of detailed health and nutrition statistics for select eligible children, in addition to characteristics about the woman. The birth history forms the basis for the Kids Recode file, a standardized module of information related to the child’s pregnancy and postnatal care and immunization, health and nutrition data [72]. The recode file is a standardized file that facilitates cross-country analysis.

We calculate anthropometric measurements, including *weight* (tenths of a kilogram) and *height* (tenths of a centimeter) from the Kids Recode files. We construct the z-sores of anthropometric indices using Stata Statistical Software 15.1 [73,74]. We input the *weight* and *height* measurements along with the *sex* and *child’s age* to calculate z-scores following the 2006 World Health Organization growth standards [10]. The Nigerian sample includes 7361 (15.3% prevalence) *wasted* and 18,723 (38.9% prevalence) *stunted* children. The Kenyan sample includes 1775 (6.3% prevalence) *wasted* and 8396 (29.5% prevalence) *stunted* children. The dependent variables, *wasting* and *stunting*, are child-level composite binary indicators equal to one if the child’s calculated z-score is below negative two standard deviations from the reference median and zero otherwise. 

Unique identifiers link the georeferenced data to records in the household surveys at the cluster level. However, the Demographic and Health Surveys employ geographic-masking with a coordinate displacement process to protect respondent confidentiality. The process displaces urban clusters up to two kilometers, displaces rural clusters up to five kilometers, and randomly selects one percent of the rural clusters to displace up to ten kilometers [71]. We link the Kids Recode files via timestamps and the cluster-level spatial identifiers to remotely monitored climatic variables (for more on variable composition, see Appendix A).

The Climate Hazards InfraRed Precipitation with Stations (CHIRPS) precipitation data product [75] and the Climate Hazards InfraRed Temperature with Stations (CHIRTS) temperature data product [76] provide interpolated high spatial resolution and coverage estimates of climatological activity between 1983 and 2016 inclusively. These products are used to estimate average total monthly rainfall, average total growing season rainfall, average maximum monthly temperature, and average maximum growing season temperature. And the normalized difference vegetation index (NDVI) is a continuous unitless *greenness index* of growing season abundance [77].

Selected covariates follow the UNICEF [29] conceptual framework along with spatially explicit temperature, precipitation, NDVI, and anomaly climatic inputs [75,76,77]. The emphasis of the model is on accommodating many possible determinants of malnutrition and prioritizing the most important within a specific contextual application while being easy to communicate across different users. Table 1, Table 2 and Table 3 report the summary statistics of discrete variables, continuous variables, and the hierarchical decomposition of DHS, respectively. 

### 2.3. Statistical Analysis

Hierarchical models respect the heterogeneity of social experience and provides appropriate generalization to account for differences across groups [78]. Each child, household, cluster, and state has its own distinctive variation and characteristics. Hierarchical modeling attenuates the risk of both ecological and atomistic fallacies by considering all levels simultaneously [79]. 

Within a hierarchical modeling structure, we specify and measure the variability associated with different units of observation levels––child, household, cluster, and state––to match the Demographic and Health Surveys data structure. We assume each level is a pure hierarchical set, such that all clusters are contained within one and only one state, all households are contained within one and only one cluster, and all children are contained within one and only one household (for more on model specification, see Appendix B). 

Variables at each level explain the measurement variability and its effect on malnutrition. Effects may also vary randomly among the units at higher levels (i.e., *cross-level variability*). For example, the magnitude of the effect of a child’s gender on their probability of being wasted may depend on cluster level characteristics, such as easy access to an improved toilet. Random variability may also exist at the household, cluster, or state scale––implying random intercepts. Explicit formulation of a hierarchical model with effects at, within, and between levels ameliorates these issues of conceptualization [80].

Inherently, child malnutrition is an individual and household-level phenomenon, yet it is at the country (and subnational) levels that many policy decisions are made. Using average data can be misleading if distribution is important and differs across countries and conclusions from cross-national data may not be applicable to individual countries’ situations [2]. Our approach addresses the interdependency explicitly. We preserve the units-of-analysis across levels in a combined structure and estimate random effects for each organizational unit. The standard error estimates incorporate the variability of the random effects and adjust for intraclass correlation [80].

Aggregation bias may occur when a variable has a different meaning, and thus a different effect, at different hierarchical levels. For example, the average quality of water and sanitation in a cluster may have an effect on a child’s health above and beyond the effect of an individual child’s water and sanitation circumstances at home. Our approach alleviates confounding effects by partitioning the effect of water and sanitation quality on health into separate components. 

Misestimating precision may occur in standard error estimates if a model fails to account for dependence among individual responses within a group. Once the grouping has been established, even if it is established at random, the group itself will tend to become differentiated [81]. The group and its members can both influence and be influenced by the composition of the group [82]. Continuing from the previous example, the survey design may have selected the survey clusters at random yet the composition of children within a cluster is likely interdependent. An individual child’s water and sanitation circumstance is reliant on the available infrastructure and cultural conventions of that child’s community and so, too, is the child living next door, but far less so is the child living five states away. 

We use the full assemblage of data across each organizational level to provide separate predicted probabilities for each category of interest. The estimators are weighted composites from the category of interest and the overall sample. *Within group* units are more similar than *between group* units and across levels, which mimics *the first law of geography*—everything is related to everything else, but near things are more related than distant things [83]. Children within the same household tend to be more similar to each other than those in other households, similarly for households within clusters, and clusters within states, and even for children within clusters, and households within states. This clustering occurs through some mechanism interconnected to unit characteristics: siblings do not end up in the same household by random chance. Generalization of classical regression methods with hierarchical methods is almost always an improvement in terms of fit, prediction and inference [84]. All results and conclusions are drawn from a *four-level random intercept hierarchical generalized logit model* specification.

## 3. Results

Figure 1 shows that the prevalence rates of wasting and stunting are overall spatially correlated, although there are pockets where rates deviate substantially suggesting different causal pathways [85]. The variable heterogeneity of malnutrition prevalence over the landscape highlights the need for a disaggregate and spatially explicit modeling approach (for more detailed spatial distributions of uncertainty estimates, see Appendix C). The Demographic and Health Surveys data form a natural hierarchical structure: regions within a country, states within a region, clusters within a state, households within a cluster, occupants within a household, and children for each woman.

The results across the various modeling approaches tell a consistent story, implying the results are robust to particular modeling variations. Our results as summarized in Table 4 indicate that the hierarchical structure alone explains 28 to 36 percent of the variation in malnutrition, meaning the additional model complexity has consequential explanatory value.

Figure 2 visualizes the results of the discrete covariates. It illustrates how much each categorical determinant affects malnutrition for a change from a baseline counterfactual. Generally, the effect sizes for stunting are larger than for wasting due to the smaller prevalence of wasting in the population. Because the model results measure the direct impact on the percentage point difference in probability of malnutrition in the population (i.e., prevalence), the size of the marginal effects have an upper-bound limit of the prevalence in the population. In other words, only already wasted and stunted children can transition to being non-wasted and non-stunted. 

### 3.1. Social Determinants

Table 5 indicates that in both Nigeria and Kenya, the mother’s education plays a greater role in determining wasting, whereas household wealth is the leading determinant of stunting. On average, in Nigeria, the probability of being wasted is 4 percentage points (95% CI: −5.4 to −2.7) lower for a child from a mother with higher education than from a mother with no education, whereas in Kenya, the probability of being wasted is 1.7 percentage points (95% CI: −2.6 to −0.89) lower for a child from a mother with higher education than from a mother with no education. That is to say, the absolute prevalence rates of wasting in Nigeria and Kenya would drop from 15.31% and 6.25%, respectively, down to 11.31% and 4.55% if mothers of wasted children had higher education holding all else constant. Table 6 confirms that education plays a vital role in stunting prevalence as well. Mothers attaining higher education are associated with a reduction in stunting rates by 13 percentage points (95% CI: −16 to −10) in Nigeria and 5.9 percentage points (95% CI: −10 to −1.3) in Kenya. In other words, education alone has the potential to curtail the number of stunted children by over one-third.

In terms of quantifying the results in numbers of children, and in numbers of deaths prevented, the effects are highly epidemiologically significant. In 2011, the Nigeria under-five population was 27,195,000 with a 41% stunting prevalence (11,149,950) and a 14% wasting prevalence (3,807,300), with an overall mortality rate of 124/1000 for under-fives (3,372,180). This rate is much lower for non-malnourished children making the estimated projections of deaths prevented a conservative lower bound of the true value [10]. Using a maximally adjusted, minimum hazard ratio, of 2.12 for stunting mortality and 3.47 for wasting mortality, the mortality rate becomes 260/1000 at a minimum for stunted children, and 430/1000 at a minimum for wasted children [86]. At a maximum education is associated with a reduced stunting prevalence of 13 percentage points, or by 3,353,350 children, thus increasing education may prevent at a minimum 490,989 deaths. Similarly, if at a maximum education is associated with a reduced wasting prevalence of 4 percentage points, or by 1,087,800 children, increasing education may prevent at a minimum 315,767 deaths.

In 2011, the Kenya under-five population was 6,805,000 with a 35% stunting prevalence (2,381,750) and a 7% wasting prevalence (476,350), with an overall mortality rate of 73/1000 for under-fives (496,765). This rate is much lower for non-malnourished children, making the deaths prevented estimates conservative lower bounds of their true values [10]. Using a maximally adjusted minimum hazard ratio of 2.12 for stunting mortality and 3.47 for wasting mortality, the mortality rate becomes 150/1000 at a minimum for stunted children and 250/1000 at a minimum for wasted children [86]. If, at a maximum, education is associated with a reduced stunting prevalence of 5.9 percentage points, or by 401,495 children, then increasing education may prevent at a minimum 32,826 deaths. Similarly, if, at a maximum, education is associated with a reduced wasting prevalence of 1.7 percentage points, or by 115,685 children, increasing education may prevent at a minimum 21,206 deaths.

Results in Table 5 and Table 6 also confirm the powerful influence of wealth on malnutrition rates. The richest families from the highest wealth quintile exhibit a reduced wasting prevalence of 0.95 percentage points (95% CI: −2.5 to 0.63) in Nigeria and 1.2 percentage points (95% CI: −2.3 to −0.17) in Kenya. Wealth is associated with a reduced stunting prevalence of 16 percentage points (95% CI: −18 to −13 and −19 to −12) in both Nigeria and Kenya. The second highest wealth quintile is associated with a reduced wasting prevalence of 1.6 percentage points (95% CI: −2.8 to −0.42) in Nigeria and 1.1 percentage points (95% CI: −1.8 to −0.3) in Kenya. Similarly it is associated with a reduced stunting prevalence of 12 percentage points (95% CI: −15 to −9.9) in Nigeria and 10 percentage points (95% CI: −13 to −6.9) in Kenya (Table 5 and Table 6). Moving to the middle wealth quintile is associated with a reduced wasting prevalence of 1.3 percentage points (95% CI: −2.2 to −0.45) in Nigeria and 0.79 percentage points (95% CI: −1.5 to −0.04) in Kenya, and a reduced stunting prevalence of 6 percentage points (95% CI: −8.2 to −3.9) in Nigeria and 8.1 percentage points (95% CI: −11 to −5.4) in Kenya (Table 5 and Table 6). Even moving from the poorest to second-poorest wealth quintile is associated with a reduced wasting prevalence of 0.06 percentage points (95% CI: −0.92 to 0.8) in Nigeria and 0.92 percentage points (95% CI: −1.6 to −0.22) in Kenya, and a reduced stunting prevalence of 2.9 percentage points (95% CI: −4.7 to −1.1) in Nigeria and 4.3 percentage points (95% CI: −6.7 to −1.9) in Kenya. Overall, changes in wealth alone have a smaller but substantial impact on wasting with reductions up to one fifth. Even more substantially, wealth alone has the potential to curtail the number of stunted children by more than one half.

### 3.2. Environmental Determinants

The climate variables are a remote monitoring corollary to malnutrition. Climate variables have the potential to act as leading indicators for changes in malnutrition prevalence with wide coverage and lower costs compared to traditional clinical survey techniques. Malnutrition is often purported to be the most significant impact of climate change on children’s health, but little empirical evidence exists in the literature [36,87,88]. For Nigeria and Kenya, NDVI, precipitation, and temperature levels all play a significant, but not homogeneous, role in determining wasting and stunting prevalence.

Figure 3 shows that the increased precipitation levels in the preceding growing season may have an ameliorative effect. Precipitation levels in Kenya reaching an average 2.5 dm during the growing season is associated with a 3% predicted prevalence of wasting: an over 50% percent reduction from the sample average, whereas a precipitation level of 6.0 dm over the growing season in Nigeria is associated with a one in ten predicted prevalence of wasting. Figure 4 presents the effect of temperature on average predicted probability of malnutrition. In the case of Nigeria, it shows that higher temperatures correspond to higher wasting prevalence. On average, a maximum monthly temperature of 38 °C in the preceding growing season is associated with a 25% wasting prevalence. That is to say, the higher temperature corresponds to a 10-percentage point higher wasting prevalence. Under a forecasting regime, the results show that temperatures in Nigeria reaching an average monthly maximum of 38 °C during the growing season is associated with one in four children experiencing wasting the following year. Similarly, an average monthly maximum temperature of 35 °C in Kenya during the growing season is associated with one in ten children experiencing wasting the following: a nearly two-fold increase from the observed prevalence.

NDVI in the preceding growing season is a further measure with a strong inverse relationship to wasting rates. Figure 5 indicates that in both Nigeria and Kenya, moving from the lowest to the highest values of observable NDVI is associated with a reduced wasting rate of 50%.

While the absolute value or level plays the largest and most direct determining role in malnutrition outcomes, the long-term variability or anomaly plays a substantial secondary role, too. As presented in Table 5 and Table 6, one standard deviation increase in precipitation anomaly is associated with a reduced wasting prevalence of 0.12 percentage points (95% CI: −1.28 to 1.05) in Nigeria and a reduced stunting prevalence of 0.5 percentage points (95% CI: −1.18 to 0.18) in Kenya. Precipitation anomaly is associated with an increased wasting prevalence of 0.16 percentage points (95% CI: −0.13 to 0.46) in Kenya and an increased stunting prevalence of 1.36 percentage points (95% CI: −0.26 to 2.88) in Nigeria. One standard deviation increase in temperature anomaly is associated with a reduced wasting prevalence of 1.24 percentage points (95% CI: −2.39 to −0.12) in Nigeria and has zero detectable effect (95% CI: −0.28 to 0.27) in Kenya. Temperature anomaly is associated with a reduced stunting prevalence of 0.83 percentage points (95% CI: −2.67 to 0.97) in Nigeria, but an increased prevalence of 0.45 percentage points (95% CI: −0.19 to 1.08) in Kenya. One standard deviation increase in NDVI anomaly is associated with an increased wasting prevalence of 0.11 percentage points (95% CI: −0.36 to 0.6) in Nigeria and 0.19 percentage points (95% CI: −0.07 to 0.44) in Kenya. NDVI anomaly is associated with an increased stunting prevalence of 0.78 percentage points (95% CI: −0.52 to 2.08) in Nigeria, but a decreased prevalence of 0.44 percentage points (95% CI: −1.22 to 0.33) in Kenya.

### 3.3. Goodness of Fit

To assess fit we calculate percent correctly classified, McIntosh–Dorfman criterion, and McFadden’s pseudo-R-squared measurements (for specifics, see Appendix D). However, we rely on a decision analysis approach to generate predicted probability cutoff values, which we use to estimate the probability of underlying malnutrition. The decision curve analysis estimates of the hierarchical model specification are on average 15.3% for Nigeria wasting; 4.5% for Kenya wasting; 38.7% for Nigeria stunting; and 31.1% for Kenya stunting. The subsequent measures of sensitivity (true-positive rate) and specificity (true-negative rate) under the maximized net benefit regime range from 77.2% at a minimum to a maximum of 95.3% with a value of 84.2% on average, indicating a good fit.

### 3.4. Limitations

Using a standardized questionnaire model, the Demographic and Health Surveys Program aims to collect data that are comparable across countries. However, the questionnaire model has been modified across each of the seven phases of the Program making it difficult to measure changes through time. Given our samples are stratified and relatively large, sampling errors of excessive skepticism are low, but remain non-zero. Nor do we consider the overall quality of the anthropometric data itself as an approximation of normally distributed z-scores standard deviations [89].

In the survey design, individuals within households are not sampled, only clusters are sampled, and then households are sampled within clusters. Given that the DHS datasets do not provide a separate sampling fraction (i.e., weights) for clusters, households, and individuals for privacy, weighting in a multilevel model is infeasible [90,91,92]. The DHS geographic displacement process reduces the risk of disclosing confidential personal information, but adds artificial uncertainty into the signal-to-noise ratio and lowers the precision of estimated covariates.

Given the availability of DHS data release cycles coupled with the need to overlap with the spatially explicit historical CHIRTS and CHIRPS climate data, the temporal window for analysis is limited. And although the data have a temporal component, successive surveys are repeated cross sections, not a panel. There remains a need for similar studies that include more countries, across more surveys, across a broader timespan, examining more outcomes with more inputs specifically directed at the nexus of climate, conflict, and malnutrition.

Given the model specification complexity we did not control for potential misclassification error in the outcome variable, which may cause attenuated coefficient estimates [93,94]. Exploratory analysis suggests the accuracy of the observed wasted children is as low as 37% (Nigeria) and 21% (Kenya). The accuracy of the observed stunted children is better, 78% (Nigeria) and 66% (Kenya). Other estimates for the overdispersion of height-for-age z-scores suggest variance inflation factors as high as 110% [95].

Accompanying the model complexity is the opportunity to mis-specify the model. For example, it does not make sense to include a district-level random slope for the variable *number of hospitals in a district* as it does not vary within the district. However, low within-cluster variance is not much of an issue at all as long as there are an adequate number of clusters that do have sufficient data [96]. We performed specification robustness analyses, including linear probability specification and logit specifications (results available upon request). Given the discrete nature of the dependent variables, wasting and stunting, we use linear probability and logit models to motivate the initial coefficient interpretations and provide a lower bound on effect sizes. Exploring multiple model specifications helps to minimizes specification error and maximizes validity, and utilizing different populations defends against confounding [97].

## 4. Discussion

Although wasting and stunting are related malnutrition indicators their causal pathways, prevalence, duration, impact, and determining factors are distinct. Across both Nigeria and Kenya, stunting is most significantly impacted by family wealth, followed by the mother’s education, a clinical delivery, vaccinations, and children who are asymptomatic of fever, cough, or diarrhea. In Nigeria diet diversity manifests as a mitigating stunting risk factor, whereas in Kenya access to improved latrine facilities and rural households mitigates stunting prevalence. Wasting is most significantly impacted by mother’s education, followed by family wealth, a clinical delivery, and vaccinations across both Nigeria and Kenya. And in Nigeria, children living in urban households and those children exhibiting symptoms of fever, cough, or diarrhea are also at elevated wasting risk levels.

One likely cause of the discrepancy is the overall sample dispersion across the two countries, coupled with the greater potential reduction for Nigeria to curb malnutrition prevalence given its higher observed rates. For example, in Nigeria, mothers’ education exhibits a more bimodal distribution, where over 45% of mothers have no formal education and over 30% have secondary or higher, whereas in Kenya, there is a more clumped distribution centered around primary education, which accounts for over 54% of the sample. Given these two very different distributions, the model results suggest that in a more divided society with a high prevalence of malnutrition, having greater access to education will afford proportionally greater gains in beneficial public health outcomes.

Climatic variables are powerful determinants of malnutrition. Across the observable range of values, changes in precipitation, temperature, or NDVI (in the preceding growing season) alone could curtail or inflate the number of wasted and stunted children by more than one half. However, their effects can vary greatly across different nutrition indicators and different countries (congruent with previous efforts elsewhere) [42,98]. Due to the distinct causal pathways and chronic nature of stunting, the signal-to-noise ratio of climate determinants is markedly diminished. In Kenya, higher precipitation and NDVI levels were deleterious and significant determinants, while higher temperature levels were a mitigating and significant determinant. Yet, in Nigeria higher temperature, precipitation, and NDVI levels were mitigating determinants.

Surprisingly, some oft-purported determinants of malnutrition were not significant in either the statistical or epidemiological sense. These include climate anomalies, access to improved latrine facilities, access to improved water facilities, weaning practices, and diet diversity for wasting. Similarly, for stunting, improved water facilities and weaning are not significant. Further research is needed to ameliorate these discrepancies. In particular, agencies and organizations such as the Famine Early Warning Systems Network, Action Against Hunger, the International Food Policy Research Institute, the World Bank, the World Food Program, UNICEF, the World Health Organization, and many others rely accurate and precise information to inform their models, forecasts, and resource deployment.

One segment of the public health community has decided that certain determinants are a public health boon. Therefore, these determinants must be drastically buoyed. The force with which these conclusions are presented is not in reasonable balance with the strength of the evidence. Programs, once in place develop a life of their own; the possibility of health benefits becomes probability, which becomes certainty. The appearance of scientific unanimity is a powerful political tool, especially when the evidence is mixed. Dissent becomes a threat, which must be marginalized. If funding agencies and journals are unwilling to brook opposition, rational discussion is curtailed. There soon comes about the pretense of policy based on scientific inquiry—without the substance.

In the clinical setting of public health and epidemiology, a diagnostic application helps to estimate the probability that malnutrition is present, identifying the nature or cause of the malnutrition; whereas a prognostic application helps to predict how malnutrition will develop and target preventive interventions to children at relatively high risk [99]. Diagnostics can be described as the probability of malnutrition conditional on a set of latent determinants, whereas prognostics can be thought of as the obverse or the probability of future outcomes conditional on being malnourished.

Estimated effects from latent determinants provide the diagnostic insights, whereas harm versus benefit establishes the prognostic framework. The purpose of a prognostic model is that better decisions are made with the model than without. Within the prognostic framework reliability of predictions is key. It is our aim to estimate the probability of malnutrition in a diagnostic sense, and to help target preventive interventions in a prognostic sense. Understanding how the distinctiveness of location effects malnutrition provides even more clarity.

## 5. Conclusions

Malnutrition devastates millions of children every year, yet the latent determinants are largely obscure. One is best informed by examining determinants on the basis of quantitative and epidemiological significance. A determinant’s impact is best measured by its ability to change malnutrition prevalence in an epidemiologically significant way. We find the most impactful latent determinants each have the capacity to reduce prevalence rates by as much as 50%: an epidemiologically significant effect.

The inconsistencies of determinants across space and malnutrition outcomes highlight the need for prudent, highly specific, and tailored approaches, especially when using climate determinants for any forecasting efforts or policy interventions [100,101]. Particular focus should be paid to those determinants that are either actionable by policy intervention or serviceable in forecasting and intervention efforts as well as epidemiologically significant. Identifying effective mitigating determinants to prevent the harmful effects of malnutrition in children should be a priority. Only with explicit identification and measurement can intervention organizations and governments begin to make substantial progress to reduce childhood malnutrition.

## Figures and Tables

**Figure 1 nutrients-16-02014-f001:**
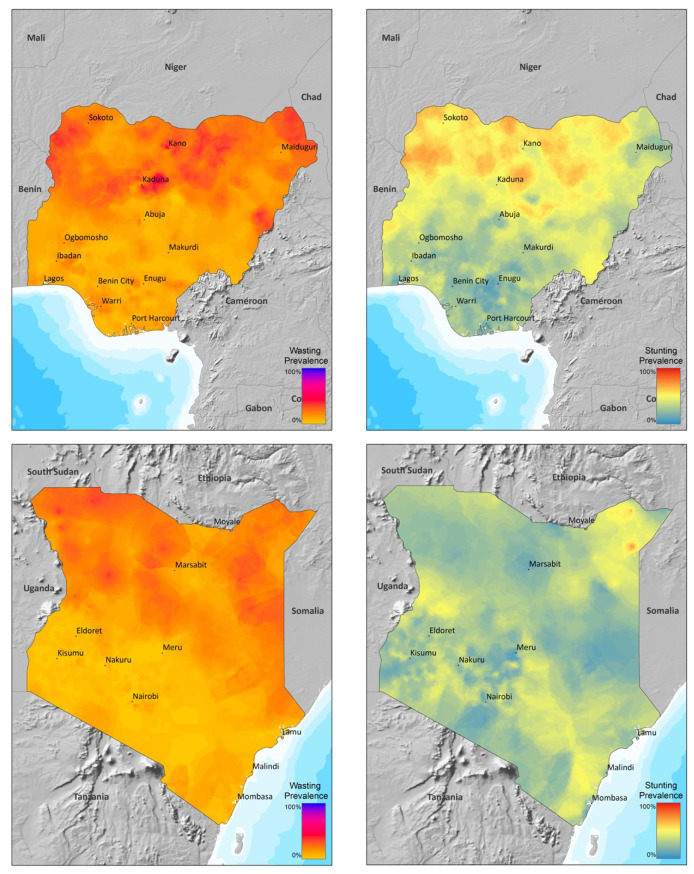
Empirical Bayesian kriging of sample malnutrition prevalence across Kenya and Nigeria DHS-IV, DHS-V, and DHS-VI using ArcGIS Desktop 10.5.1 software by ESRI. Color gradients indicate prevalence of stunting and wasting malnutrition rates.

**Figure 2 nutrients-16-02014-f002:**
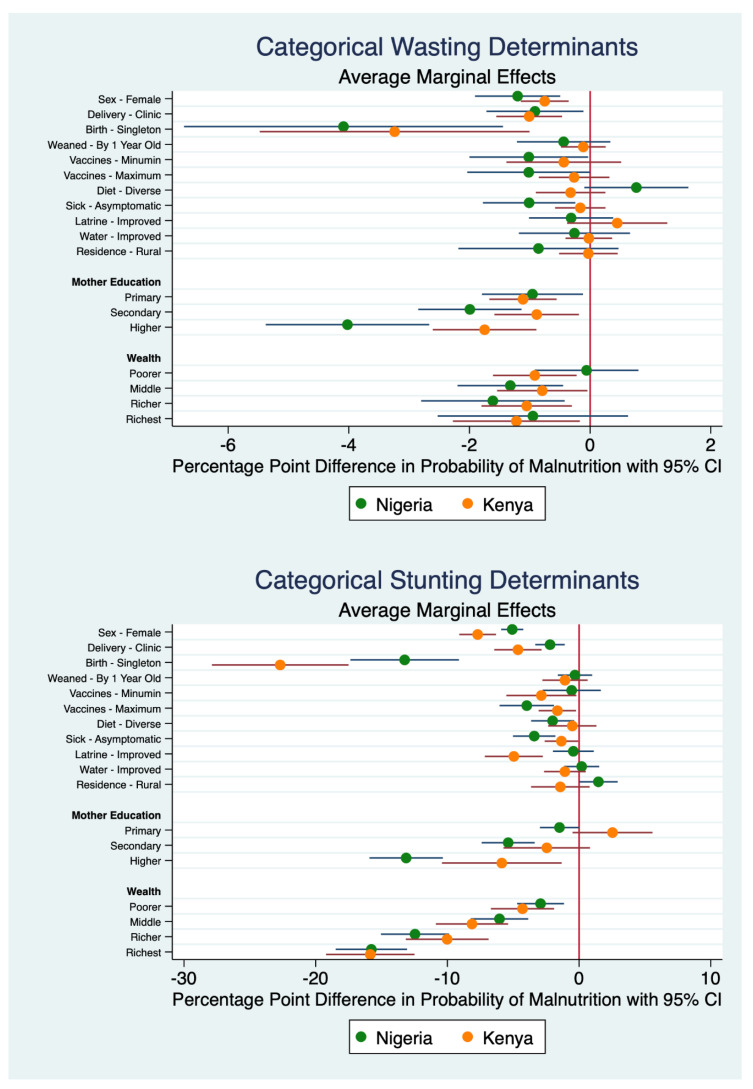
Average marginal effects of categorical determinants of malnutrition (based on Table 5 and Table 6). Variables are displayed such that negative values are beneficial for children’s health and positive values are deleterious for children’s health. The vertical red line at zero marks the liminal threshold, whereas the green and orange horizontal lines are 95% confidence intervals.

**Figure 3 nutrients-16-02014-f003:**
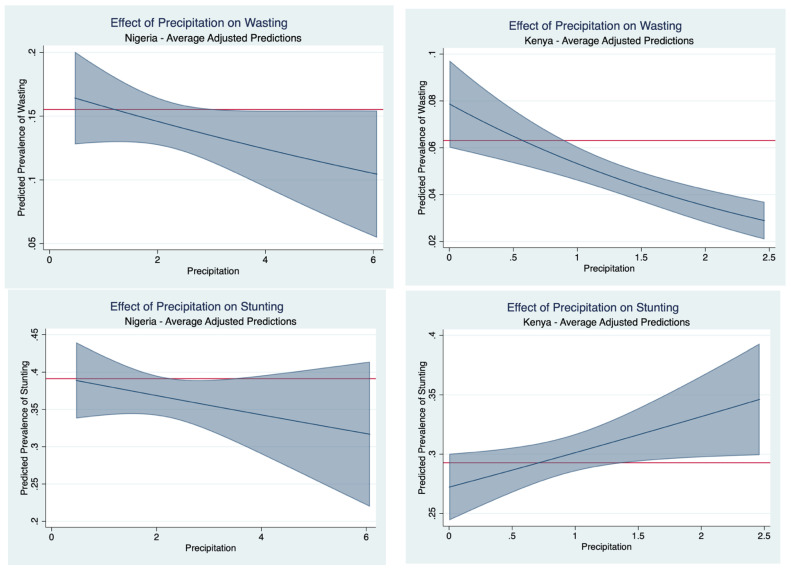
Effect of precipitation on average predicted probability of malnutrition. The horizontal axis is the in-sample range of average total monthly rainfall (dm) during the preceding growing season. The horizontal red line demarks the observed malnutrition prevalence and the sloped blue line illustrates how much the expected prevalence rates change as precipitation changes. The shaded blue corresponds to a 95% confidence interval band on the estimate.

**Figure 4 nutrients-16-02014-f004:**
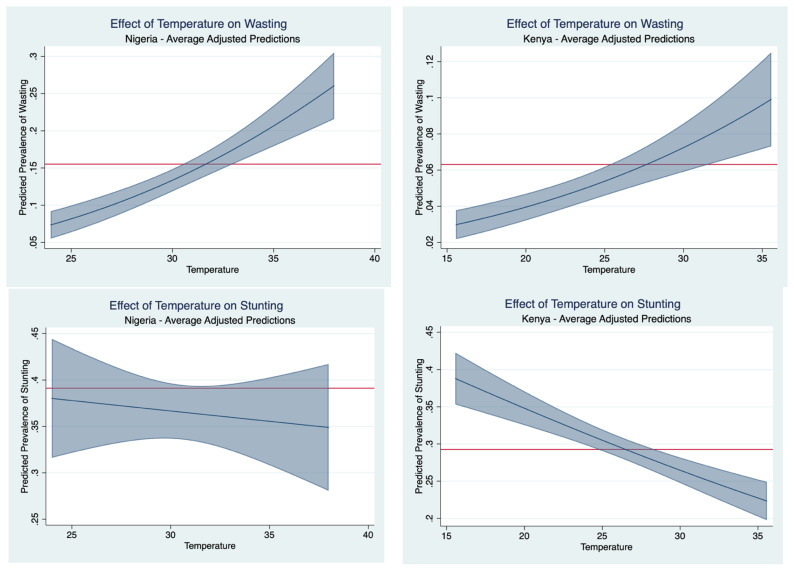
Effect of temperature on average predicted probability of malnutrition. The horizontal axis is the in-sample range of average maximum monthly temperatures (°C) during the preceding growing season. The horizontal red line demarks the observed malnutrition prevalence and the sloped blue line illustrates how much the expected prevalence rates change as temperature changes. The shaded blue corresponds to a 95% confidence interval band on the estimate.

**Figure 5 nutrients-16-02014-f005:**
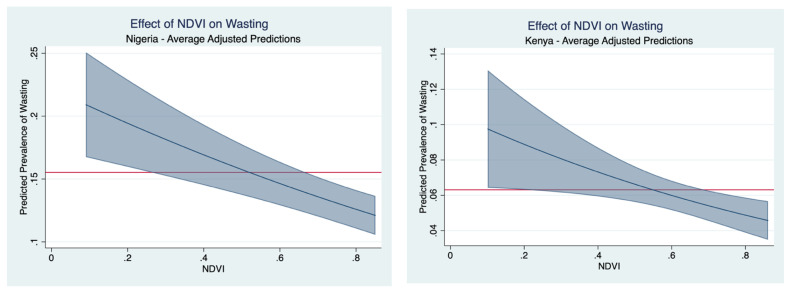
Effect of NDVI on average predicted probability of malnutrition. The horizontal axis is the in-sample range of the unitless NDVI for the three greenest months during the preceding growing season. The horizontal red line demarks the observed malnutrition prevalence and the sloped blue line illustrates how much the expected prevalence rates change as NDVI changes. The shaded blue corresponds to a 95% confidence interval band on the estimate.

**Table 1 nutrients-16-02014-t001:** Summary statistics of discrete variables.

	Nigeria		Kenya	
*Variable*	*Frequency*	*Percent*	*Frequency*	*Percent*
**Wasting Status**				
Not wasted	40,716	84.69	26,646	93.75
Wasted	7360	15.31	1775	6.25
**Stunting Status**				
Not stunted	29,353	61.06	20,025	70.46
Stunted	18,723	38.94	8396	29.54
**Sex**				
Male	23,991	49.90	14,369	50.56
Female	24,085	50.10	14,052	49.44
**Delivery**				
Home	29,850	62.38	14,069	49.63
Clinic	18,002	37.62	14,277	50.37
**Birth**				
Multiple	1428	2.97	734	2.58
Singleton	46,648	97.03	27,687	97.42
**Weaned**				
Breastfed beyond 1 year	16,809	34.96	7158	25.19
Weaned by 1 year	19,645	40.86	14,896	52.41
Breastfed up to 1 year	11,038	22.96	4170	14.67
Weaned before 1 year	584	1.21	2197	7.73
**Vaccines—Minimum**				
No	12,181	25.36	1341	4.72
Yes	35,850	74.64	27,073	95.28
**Vaccines—Maximum**				
No	40,684	84.70	16,965	59.71
Yes	7347	15.30	11,449	40.29
**Diet**				
Unvaried	35,622	74.10	22,723	79.95
Diverse	12,454	25.90	5698	20.05
**Sick**				
Symptomatic	12,709	26.66	14,226	50.14
Asymptomatic	34,957	73.34	14,149	49.86
**Latrine—Improved**				
No	32,967	70.96	22,184	82.36
Yes	13,489	29.04	4751	17.64
**Water—Improved**				
No	22,082	47.07	11,540	41.37
Yes	24,833	52.93	16,355	58.63
**Residence**				
Urban	15,680	32.62	8179	28.78
Rural	32,396	67.38	20,242	71.22
**Mother’s Education**				
None	21,919	45.59	5992	21.08
Primary	10,898	22.67	15,521	54.61
Secondary	12,471	25.94	5280	18.58
Higher	2788	5.80	1628	5.73
**Wealth Index**				
Poorest	10,697	22.25	9077	31.94
Poorer	10,813	22.49	5784	20.35
Middle	9678	20.13	4856	17.09
Richer	9035	18.79	4333	15.25
Richest	7853	16.33	4371	15.38
**Interview Month**				
January	0	0.00	1530	5.38
February	1370	2.85	1265	4.45
March	7315	15.22	25	0.09
April	8166	16.99	729	2.57
May	8709	18.12	4042	14.22
June	3932	8.18	4718	16.60
July	6327	13.16	4828	16.99
August	5698	11.85	4035	14.20
September	4043	8.41	4163	14.65
October	2485	5.17	805	2.83
November	31	0.06	1145	4.03
December	0	0.00	1136	4.00
**Survey Phase**				
DHS-IV	4386	9.12	4718	16.60
DHS-V	19,246	40.02	5101	17.95
DHS-VI	24,454	50.85	18,602	65.45

**Table 2 nutrients-16-02014-t002:** Summary statistics of continuous variables.

	Nigeria				Kenya			
		*Standard*				*Standard*		
*Variable*	*Average*	*Deviation*	*Min*	*Max*	*Average*	*Deviation*	*Min*	*Max*
Child’s Age (Months)	28.3	17.2	0	59	28.9	17	0	59
Mother’s Age (Years)	29.5	6.93	15	49	28.6	6.57	15	49
Birth Tally	4.3	2.58	1	18	3.8	2.36	1	16
Precipitation (dm)	21.3	7.95	4.7	61.6	8.3	6.13	0.02	25.2
Temperature (°C)	31	2.23	24	38.3	26.4	3.7	15.6	35.6
Precipitation Anomaly	0.2	2.62	−11.3	11.4	−0.5	1.47	−5.5	8.2
Temperature Anomaly	−0.7	0.46	−1.9	0.7	−0.8	0.45	−2.6	0.9
NDVI	0.6	0.14	0.09	0.9	0.6	0.14	0.1	0.9
NDVI Anomaly	0.0	0.026	−0.1	0.2	0.0	0.034	−0.1	0.2

**Table 3 nutrients-16-02014-t003:** Hierarchical decomposition of DHS.

	Nigeria				Kenya			
		*Observations per Group*				*Observations per Group*		
*Scale*	*Groups*	*Min*	*Average*	*Max*	*Groups*	*Min*	*Average*	*Max*
State	37	765	1299.1	2750	47	339	600.9	1165
Cluster	2131	1	22.6	79	2365	1	11.9	43
Household	30,904	1	1.6	8	20,048	1	1.4	6
Child	48,068				28,241			

**Table 4 nutrients-16-02014-t004:** Unconditional hierarchical model—variance decomposition.

	Wasted		Stunted	
Hierarchical Fully Unconditional	Nigeria	Kenya	Nigeria	Kenya
*Variance Decomposition—Percent by Level*				
States	7.09%	11.35%	10.94%	1.87%
Clusters	9.48%	6.35%	6.99%	6.11%
Households	17.50%	20.09%	13.31%	20.08%
Children	65.93%	62.22%	68.77%	71.94%

**Table 5 nutrients-16-02014-t005:** Interpreted hierarchical analyses, wasted percentage point change.

Interpreted Results	Percent Change in Wasted Probability			
Hierarchical Random Intercept	Nigeria		Kenya	
** *For a Change from Baseline Category with 95% Confidence Interval in Brackets* **				
Sex—Female	−1.2%	[−1.9, −0.49]	−0.75%	[−1.1, −0.36]
Delivery—Clinic	−0.91%	[−1.7, −0.11]	−1%	[−1.6, −0.46]
Birth—Singleton	−4.1%	[−6.7, −1.4]	−3.2%	[−5.5, −1]
Weaned—By 1 Year Old	−0.44%	[−1.2, 0.34]	−0.11%	[−0.48, 0.26]
Vaccines—Minimum	−1%	[−2, −0.03]	−0.44%	[−1.4, 0.52]
Vaccines—Maximum	−1%	[−2, 0]	−0.27%	[−0.85, 0.32]
Diet—Diverse	0.77%	[−0.1, 1.6]	−0.32%	[−0.9, 0.25]
Sick—Asymptomatic	−1%	[−1.8, −0.25]	−0.16%	[−0.58, 0.26]
Latrine—Improved	−0.31%	[−1, 0.38]	0.45%	[−0.38, 1.3]
Water—Improved	−0.26%	[−1.2, 0.66]	−0.02%	[−0.41, 0.37]
Residence—Rural	−0.86%	[−2.2, 0.47]	−0.03%	[−0.51, 0.46]
Mothers Education				
Primary	−0.96%	[−1.8, −0.12]	−1.1%	[−1.7, −0.56]
Secondary	−2%	[−2.8, −1.1]	−0.89%	[−1.6, −0.18]
Higher	−4%	[−5.4, −2.7]	−1.7%	[−2.6, −0.89]
Wealth Index				
Poorer	−0.06%	[−0.92, 0.8]	−0.92%	[−1.6, −0.22]
Middle	−1.3%	[−2.2, −0.45]	−0.79%	[−1.5, −0.04]
Richer	−1.6%	[−2.8, −0.42]	−1.1%	[−1.8, −0.3]
Richest	−0.95%	[−2.5, 0.63]	−1.2%	[−2.3, −0.17]
** *For a 1-Unit Increase in Determinant with 95% Confidence Interval in Brackets* **				
Child’s Age	−2.2%	[−2.8, −1.5]	−0.13%	[−0.38, 0.12]
Mother’s Age	0.26%	[−0.64, 1.2]	−0.22%	[−0.65, 0.2]
Birth Tally	−0.17%	[−0.39, 0.05]	0.07%	[−0.07, 0.21]
Precipitation	−0.96%	[−2.3, 0.41]	−1.5%	[−2.5, −0.63]
Temperature	1.2%	[0.79, 1.5]	0.24%	[0.12, 0.36]
Precipitation Anomaly	−0.45%	[−4.9, 4]	1.1%	[−0.88, 3.1]
Temperature Anomaly	−2.7%	[−5.2, −0.26]	−0.01%	[−0.62, 0.61]
NDVI	−9.2%	[−14, −4.9]	−3.9%	[−6.6, −1.3]
NDVI Anomaly	4.4%	[−14, 23]	5.5%	[−2.1, 13]
** *For a 1-Standard Deviation Increase in Determinant with 95% Confidence Interval in Brackets* **				
Child’s Age	−3.15%	[−4.01, −2.15]	−0.18%	[−0.54, 0.17]
Mother’s Age	0.18%	[−0.44, 0.83]	−0.14%	[−0.43, 0.13]
Birth Tally	−0.44%	[−1.01, 0.14]	0.16%	[−0.17, 0.5]
Precipitation	−0.76%	[−1.83, 0.33]	−0.92%	[−1.53, −0.39]
Temperature	2.68%	[1.76, 3.35]	0.89%	[0.44, 1.33]
Precipitation Anomaly	−0.12%	[−1.28, 1.05]	0.16%	[−0.13, 0.46]
Temperature Anomaly	−1.24%	[−2.39, −0.12]	0%	[−0.28, 0.27]
NDVI	−1.29%	[−1.96, −0.69]	−0.55%	[−0.92, −0.18]
NDVI Anomaly	0.11%	[−0.36, 0.6]	0.19%	[−0.07, 0.44]
** *For a Sample Maximum Increase in Determinant with 95% Confidence Interval in Brackets* **				
Child’s Age	−10.82%	[−13.77, −7.38]	−0.64%	[−1.87, 0.59]
Mother’s Age	0.88%	[−2.18, 4.08]	−0.75%	[−2.21, 0.68]
Birth Tally	−2.89%	[−6.63, 0.92]	1.04%	[−1.08, 3.15]
Precipitation	−5.46%	[−13.09, 2.33]	−3.78%	[−6.3, −1.59]
Temperature	17.16%	[11.3, 21.45]	4.8%	[2.4, 7.2]
Precipitation Anomaly	−1.02%	[−11.12, 9.08]	1.51%	[−1.21, 4.25]
Temperature Anomaly	−7.02%	[−13.52, −0.68]	−0.02%	[−2.17, 2.14]
NDVI	−7.45%	[−11.34, −3.97]	−3.12%	[−5.28, −1.04]
NDVI Anomaly	1.32%	[−4.2, 6.9]	1.65%	[−0.63, 3.9]

**Table 6 nutrients-16-02014-t006:** Interpreted Hierarchical Analyses, Stunted Percentage Point Change.

Interpreted Results	Percent Change in Stunted Probability			
Hierarchical Random Intercept	Nigeria		Kenya	
** *For a Change from Baseline Category with 95% Confidence Interval in Brackets* **				
Sex—Female	−5.1%	[−5.9, −4.2]	−7.7%	[−9.1, −6.3]
Delivery—Clinic	−2.2%	[−3.3, −1.1]	−4.6%	[−6.4, −2.8]
Birth—Singleton	−13%	[−17, −9.1]	−23%	[−28, −18]
Weaned—By 1 Year Old	−0.31%	[−1.6, 1]	−1.1%	[−2.8, 0.65]
Vaccines—Minimum	−0.56%	[−2.8, 1.7]	−2.9%	[−5.5, −0.2]
Vaccines—Maximum	−4%	[−6, −1.9]	−1.6%	[−3.1, −0.22]
Diet—Diverse	−2%	[−3.6, −0.37]	−0.51%	[−2.3, 1.3]
Sick—Asymptomatic	−3.4%	[−5, −1.8]	−1.3%	[−2.6, −0.07]
Latrine—Improved	−0.43%	[−2, 1.1]	−5%	[−7.2, −2.8]
Water—Improved	0.2%	[−1.1, 1.5]	−1.1%	[−2.7, 0.51]
Residence—Rural	1.5%	[0.01, 2.9]	−1.4%	[−3.6, 0.8]
Mothers Education				
Primary	−1.5%	[−3, −0.01]	2.5%	[−0.5, 5.6]
Secondary	−5.4%	[−7.4, −3.4]	−2.5%	[−5.7, 0.84]
Higher	−13%	[−16, −10]	−5.9%	[−10, −1.3]
Wealth Index				
Poorer	−2.9%	[−4.7, −1.1]	−4.3%	[−6.7, −1.9]
Middle	−6%	[−8.2, −3.9]	−8.1%	[−11, −5.4]
Richer	−12%	[−15, −9.9]	−10%	[−13, −6.9]
Richest	−16%	[−18, −13]	−16%	[−19, −12]
** *For a 1-Unit Increase in Determinant with 95% Confidence Interval in Brackets* **				
Child’s Age	−0.75%	[−1.7, 0.16]	−2.6%	[−3.3, −1.8]
Mother’s Age	−3.6%	[−4.7, −2.5]	−4.6%	[−6.1, −3.1]
Birth Tally	0.37%	[0.06, 0.68]	1.1%	[0.67, 1.6]
Precipitation	−1.5%	[−4.4, 1.4]	3.3%	[0.33, 6.3]
Temperature	−0.26%	[−1.3, 0.73]	−0.92%	[−1.2, −0.61]
Precipitation Anomaly	5.2%	[−1, 11]	−3.4%	[−8, 1.2]
Temperature Anomaly	−1.8%	[−5.8, 2.1]	1%	[−0.43, 2.4]
NDVI	−6.6%	[−19, 6.1]	12%	[5.7, 18]
NDVI Anomaly	30%	[−20, 80]	−13%	[−36, 9.8]
** *For a 1-Standard Deviation Increase in Determinant with 95% Confidence Interval in Brackets* **				
Child’s Age	−1.08%	[−2.44, 0.23]	−3.68%	[−4.68, −2.55]
Mother’s Age	−2.49%	[−3.26, −1.73]	−3.02%	[−4.01, −2.04]
Birth Tally	0.95%	[0.16, 1.75]	2.6%	[1.58, 3.78]
Precipitation	−1.19%	[−3.5, 1.11]	2.02%	[0.2, 3.86]
Temperature	−0.58%	[−2.9, 1.63]	−3.4%	[−4.44, −2.26]
Precipitation Anomaly	1.36%	[−0.26, 2.88]	−0.5%	[−1.18, 0.18]
Temperature Anomaly	−0.83%	[−2.67, 0.97]	0.45%	[−0.19, 1.08]
NDVI	−0.92%	[−2.66, 0.85]	1.68%	[0.8, 2.52]
NDVI Anomaly	0.78%	[−0.52, 2.08]	−0.44%	[−1.22, 0.33]
** *For a Sample Maximum Increase in Determinant with 95% Confidence Interval in Brackets* **				
Child’s Age	−3.69%	[−8.36, 0.79]	−12.78%	[−16.23, −8.85]
Mother’s Age	−12.24%	[−15.98, −8.5]	−15.64%	[−20.74, −10.54]
Birth Tally	6.29%	[1.05, 11.56]	16.5%	[10.05, 24]
Precipitation	−8.54%	[−25.04, 7.97]	8.31%	[0.83, 15.86]
Temperature	−3.72%	[−18.59, 10.44]	−18.4%	[−24, −12.2]
Precipitation Anomaly	11.8%	[−2.27, 24.97]	−4.66%	[−10.96, 1.64]
Temperature Anomaly	−4.68%	[−15.08, 5.46]	3.5%	[−1.51, 8.4]
NDVI	−5.35%	[−15.39, 4.94]	9.6%	[4.56, 14.4]
NDVI Anomaly	9%	[−6, 24]	−3.9%	[−10.8, 2.94]

## Data Availability

No new data were created for this study. Restrictions apply to the availability of some of these data. Data were obtained from The Demographic and Health Surveys available at https://www.dhsprogram.com/Data/ (accessed on 25 September 2017) and from the Climate Hazards Center available at https://www.chc.ucsb.edu/data (accessed on 25 September 2017) with the permission of the providers.

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
