# Peer review of "The Socio-Environmental Determinants of Childhood Malnutrition: A Spatial and Hierarchical Analysis"

_nutrients, 2024, doi:10.3390/nu16132014_

Round 1

Reviewer 1 Report

Comments and Suggestions for Authors

Manuscript “The Socio-environmental Determinants of Childhood Malnutrition: A Spatial and Hierarchical Analysis” can be considered for publication in Nutrients after major corrections: Technical issues:

• Check manuscript for punctuation errors i.e. points after the references not before;

• Some tables and figures are not correctly fitted to template; Merit issues:

• Why Nigeria and Kenya have been chosen for this analysis? Because of geographical placement? Wealth level? Climate differences? – supplement introduction;

• Surveys’ data presented in the manuscript were collected in years 2003-2014 – these are old data; in 2024 they are possibly, almost surely, outdated. Why authors used these data in their manuscript as they not reflecting current situation in these countries? -explain and make it more visible in the text;

• In the analysis of wasting and stunting and climate’s variables (temperature, rain, NDVI, anomalies) authors did not mentioned from where they obtained data concerning these variables and from what period of time- supplement data for correct analysis; explain what is the meaning of this analysis for nutritional status of children;

• Why mothers’ education has a different value in child’s nutritional status in Nigeria vs. Kenya? The value of mothers education for children welfare also nutritional status itself is already well known and well described, but what is the source of the difference of impact of mothers education in these countries? – supplement the discussion;

• References: some references are very outdated (Clinical Nutrition from 1963 year?), there are only 3 references from 2020 and ”younger”, but many from 80’ and 90 of XX century. I can imagine that some of those papers are “classic” in the subject of the study, but this branch of science develops very fast and I’m sure authors would find some more updated articles published in last ten years;

Author Response

Manuscript “The Socio-environmental Determinants of Childhood Malnutrition: A Spatial and Hierarchical Analysis” can be considered for publication in Nutrients after major corrections:

Technical issues:

  • Check manuscript for punctuation errors i.e. points after the references not before;

RESPONSE: We agree and appreciate the reviewer’s attention to detail. We have corrected these errors throughout the paper.

  • Some tables and figures are not correctly fitted to template;

RESPONSE: We have tried our best to address this issue with limited success. We hope that this technical formatting issue can be best addressed by the journal’s production team.

Merit issues:

  • Why Nigeria and Kenya have been chosen for this analysis? Because of geographical placement? Wealth level? Climate differences? – supplement introduction;

RESPONSE: We have rewritten lines 88-96 in original manuscript to explain why Kenya and Nigeria were selected. This paragraph reads now as follows (lines 88-99).

"Globally Nigeria has the second highest number of stunted children behind India. There are over 4.8 million wasted children and over 10 million stunted children in Nigeria. And Kenya provides a measure of external validity and contrast to this analysis, with over 278 thousand wasted children and over 1.8 million stunted children [10]. Including both countries adds variability in terms of malnutrition prevalence rates, governance, climate, population, economy, and culture. Kenya and Nigeria both have available health survey data with extensive temporal and spatial coverage that overlaps with the available geolocated historical climate data. Despite similar efforts in the literature in both Kenya [39, 62-64] and Nigeria [11, 12, 65-68], this paper aims to be the first study to identify and quantify the non-illness-related and climatic determinants of stunting and wasting across Kenya and Nigeria through a spatially explicit hierarchical modeling approach consistent with the UNICEF [29, 69] conceptual framework."

  • Surveys’ data presented in the manuscript were collected in years 2003-2014 – these are old data; in 2024 they are possibly, almost surely, outdated. Why authors used these data in their manuscript as they not reflecting current situation in these countries? -explain and make it more visible in the text;

RESPONSE: In the new and revised Data Measures section, we address the issues of data age and availability more clearly. And we have expanded the Limitations section to better explain our modeling choices and data limitations. The key reason is that (lines 408-410): “Given the availability of DHS data release cycles coupled with the need to overlap with the spatially explicit historical CHIRTS and CHIRPS climate data, the temporal window for analysis is limited.”

  • In the analysis of wasting and stunting and climate’s variables (temperature, rain, NDVI, anomalies) authors did not mentioned from where they obtained data concerning these variables and from what period of time- supplement data for correct analysis; explain what is the meaning of this analysis for nutritional status of children;

RESPONSE: In new section 2.2 Data Measures, we provide a clearer description of the temperature and precipitation data products. And more technical details on variables and their relationships are presented in the Appendix. Furthermore, we have revised the discussion of climate variables, their impact, and history in the literature in the introduction section too. The new information presented in Section 2.2 is as follows:

“The Climate Hazards InfraRed Precipitation with Stations (CHIRPS) precipitation data product [75], and the Climate Hazards InfraRed Temperature with Stations (CHIRTS) temperature data product [76], provide interpolated high spatial resolution and coverage estimates of climatological activity between 1983 and 2016 inclusively. These products are used to estimate average total monthly rainfall, average total growing season rainfall, average maximum monthly temperature, and average maximum growing season temperature. And the Normalized Difference Vegetation Index (NDVI) is a continuous unitless greenness index of growing season abundance [77].

Selected covariates follow the UNICEF [29] conceptual framework along with spatially explicit temperature, precipitation, NDVI, and anomaly climatic inputs [75-77]. The emphasis of the model is on accommodating many possible determinants of mal-nutrition and prioritizing the most important within a specific contextual application while being easy to communicate across different users. Tables 1, 2, and 3 report the summary statistics of discrete variables, continuous variables, and the hierarchical de-composition of DHS, respectively.”

  • Why mothers’ education has a different value in child’s nutritional status in Nigeria vs. Kenya? The value of mothers education for children welfare also nutritional status itself is already well known and well described, but what is the source of the difference of impact of mothers education in these countries? – supplement the discussion;

RESPONSE: We have revised the discussion section to include a clearer and more in-depth discussion of potential sources of heterogeneity (lines 449-457):

“One likely cause of these discrepancy is the overall sample dispersion across the two countries, coupled with the greater potential reduction for Nigeria to curb malnutrition prevalence given its higher observed rates. For example, in Nigeria mothers’ education exhibits a more bimodal distribution where over 45% of mothers have no formal education and over 30% have Secondary or higher.  Whereas, in Kenya, there is a more clumped distribution centered around Primary education which accounts for over 54% of the sample. Given these two very different distributions, the model results suggest that in a more divided society with a high prevalence of malnutrition, having greater access to education will afford proportionally greater gains in beneficial public health outcomes.”

  • References: some references are very outdated (Clinical Nutrition from 1963 year?), there are only 3 references from 2020 and ”younger”, but many from 80’ and 90 of XX century. I can imagine that some of those papers are “classic” in the subject of the study, but this branch of science develops very fast and I’m sure authors would find some more updated articles published in last ten years;

RESPONSE: The revised manuscript now includes 61 references from 2010 onwards. And we have revised the Introduction to include almost all disaggregate nationally representative empirical studies of the determinants of childhood malnutrition in Kenya and Nigeria.

Reviewer 2 Report

Comments and Suggestions for Authors

The manuscript titled "The Socio-environmental Determinants of Childhood Malnutrition: A Spatial and Hierarchical Analysis" by Sandler and Sun provides an in-depth analysis of childhood malnutrition in Nigeria and Kenya using a hierarchical modeling approach. The study aims to quantify the non-illness-related determinants of stunting and wasting, integrating data from Demographic Health Surveys (DHS) and spatially explicit environmental data. The paper addresses a critical public health issue and contributes to the understanding of malnutrition determinants through rigorous statistical analysis.

The introduction effectively sets the stage for the study by highlighting the global burden of childhood malnutrition and the need for more targeted interventions.

The methodology is well-detailed, explaining the hierarchical structure of the data and the modeling approach. However, the manuscript could benefit from a more detailed explanation of how the climatic variables were integrated into the model and any potential limitations related to these data sources

Please, consider to split the paragraph: "2.1. Data Measures and Study Design" in at least 2 subparagraph in order to increase readability.

Please, revise lines 169-170 there are some issues with the references (just as an example. Many other sections of the manuscript have the same issue).

lines 158-162 there is a repetition of the same sentence 

The discussion on the varying impacts of climatic variables across different regions and indicators adds depth to the findings. However, the manuscript could further explore the implications of these variations for policy interventions

The conclusion effectively summarizes the key findings and suggests directions for future research​

  •  
Comments on the Quality of English Language

The manuscript is generally well-written, but a thorough proofread is recommended to correct minor grammatical errors and improve readability.

Author Response

The manuscript titled "The Socio-environmental Determinants of Childhood Malnutrition: A Spatial and Hierarchical Analysis" by Sandler and Sun provides an in-depth analysis of childhood malnutrition in Nigeria and Kenya using a hierarchical modeling approach. The study aims to quantify the non-illness-related determinants of stunting and wasting, integrating data from Demographic Health Surveys (DHS) and spatially explicit environmental data. The paper addresses a critical public health issue and contributes to the understanding of malnutrition determinants through rigorous statistical analysis.

  • The introduction effectively sets the stage for the study by highlighting the global burden of childhood malnutrition and the need for more targeted interventions.

RESPONSE: We agree and appreciate the reviewer’s endorsement.

  • The methodology is well-detailed, explaining the hierarchical structure of the data and the modeling approach. However, the manuscript could benefit from a more detailed explanation of how the climatic variables were integrated into the model and any potential limitations related to these data sources

RESPONSE: In new section 2.2 Data Measures, we provide a clearer description of the temperature and precipitation data products. And more technical details on variables and their relationships are presented in the Appendix. Furthermore, we have revised the discussion of climate variables, their impact, and history in the literature in the introduction section too. The new information presented in Section 2.2 is as follows:

“The Climate Hazards InfraRed Precipitation with Stations (CHIRPS) precipitation data product [75], and the Climate Hazards InfraRed Temperature with Stations (CHIRTS) temperature data product [76], provide interpolated high spatial resolution and coverage estimates of climatological activity between 1983 and 2016 inclusively. These products are used to estimate average total monthly rainfall, average total growing season rainfall, average maximum monthly temperature, and average maximum growing season temperature. And the Normalized Difference Vegetation Index (NDVI) is a continuous unitless greenness index of growing season abundance [77].

Selected covariates follow the UNICEF [29] conceptual framework along with spatially explicit temperature, precipitation, NDVI, and anomaly climatic inputs [75-77]. The emphasis of the model is on accommodating many possible determinants of mal-nutrition and prioritizing the most important within a specific contextual application while being easy to communicate across different users. Tables 1, 2, and 3 report the summary statistics of discrete variables, continuous variables, and the hierarchical de-composition of DHS, respectively.”

Further related revision can be found in the Introduction, Limitations, and Discussion sections.

  • Please, consider to split the paragraph: "2.1. Data Measures and Study Design" in at least 2 subparagraphs in order to increase readability.

RESPONSE: The former subsection 2.1 has been subdivided into “2.1 Health Survey Study Design” and “2.2 Data Measures” for better readability and content cohesion.

  • Please, revise lines 169-170 there are some issues with the references (just as an example. Many other sections of the manuscript have the same issue).

RESPONSE: All references to Tables and Figures are revised to avoid confusion and repeating. For example, the revised lines 169-170 in the original manuscript now reads as follows (lines 175-177):

“Tables 1, 2, and 3 report the summary statistics of discrete variables, continuous variables, and the hierarchical decomposition of DHS, respectively.”

  • lines 158-162 there is a repetition of the same sentence

RESPONSE: These lines have been replaced with a clearer description of the temperature and precipitation data products (lines 163-170):

“The Climate Hazards InfraRed Precipitation with Stations (CHIRPS) precipitation data product [75], and the Climate Hazards InfraRed Temperature with Stations (CHIRTS) temperature data product [76], provide interpolated high spatial resolution and coverage estimates of climatological activity between 1983 and 2016 inclusively. These products are used to estimate average total monthly rainfall, average total growing season rainfall, average maximum monthly temperature, and average maximum growing season temperature. And the Normalized Difference Vegetation Index (NDVI) is a continuous unitless greenness index of growing season abundance [77].”

  • The discussion on the varying impacts of climatic variables across different regions and indicators adds depth to the findings. However, the manuscript could further explore the implications of these variations for policy interventions

RESPONSE: We have revised the discussion section to include a clearer and more in-depth discussion of policy intervention implications. We highlight that “In particular, agencies and organizations such as the Famine Early Warning Systems Network, Action Against Hunger, the International Food Policy Research Institute, the World Bank, the World Food Program, UNICEF, the World Health Organization and many others rely accurate and precise information to inform their models, forecasts, and resource deployment.”

  • The conclusion effectively summarizes the key findings and suggests directions for future research.

RESPONSE: We agree and appreciate the reviewer’s endorsement.

Comments on the Quality of English Language

  • The manuscript is generally well-written, but a thorough proofread is recommended to correct minor grammatical errors and improve readability.

RESPONSE: We have done a thorough proofread to resolve many minor issues and improve readability.